# Impact of the Paramedic Role on Athlete Care, Emergency Response, and Injury Prevention in Sports Medicine: A Scoping Review

**DOI:** 10.3390/healthcare13182301

**Published:** 2025-09-14

**Authors:** Yasir Almukhlifi, Maher Alsulami, Adnan Alsulami, Nawaf A. Albaqami, Abdulrahmn M. Bahmaid, Salman A. Aldriweesh, Sharifah Albounagh, Krzysztof Goniewicz

**Affiliations:** 1College of Applied Medical Sciences, King Saud Bin Abdulaziz University for Health Sciences, Jeddah 22384, Saudi Arabia; alsulamim@ksau-hs.edu.sa (M.A.); sulamiad@ksau-hs.edu.sa (A.A.); 2King Abdullah International Medical Research Center, Jeddah 21423, Saudi Arabia; 3Department of Emergency Medical Services, Prince Sultan Bin Abdulaziz College for Emergency Medical Services, King Saud University, Riyadh 11451, Saudi Arabia; naalbaqami@ksu.edu.sa; 4Emergency Medical Services Team, Sports Medicine Department, MAHD Sports Academy, Riyadh 12752, Saudi Arabia; ambahmaid@mahdacademy.sa (A.M.B.); saldriweesh@inaya.edu.sa (S.A.A.);; 5Department of Emergency Medical Services and Critical Care, College of Applied Medical Sciences, Inaya Medical Colleges, Riyadh 13541, Saudi Arabia; 6Department of Security, Polish Air Force University, 08-521 Dęblin, Poland

**Keywords:** emergency medical services, sports medicine, athletic injuries, first aid, paramedics, wounds and injuries prevention & control

## Abstract

**Introduction:** Paramedics are increasingly being recognized as essential contributors to sports medicine, where their role extends beyond emergency response to prevention, planning, and collaboration with other medical professionals. Yet their scope of practice and effectiveness across sporting levels and regions remain insufficiently synthesized. **Methods:** This scoping review mapped international evidence on paramedics in sports medicine. Literature published in English between 2013 and 2023 was systematically searched in PubMed, Scopus, and ScienceDirect, and eligible studies were analyzed thematically. Thirty studies were included, spanning professional and amateur sports in North America, Europe, Asia, Oceania, and Africa. **Results:** The findings demonstrate that paramedics provide critical value across six domains. First, rapid emergency response, supported by innovations such as motorcycle-based ambulances, significantly reduced access times and improved survival rates. Second, preparedness and ongoing training, including physical fitness and interprofessional education, were shown to enhance effectiveness in demanding sporting environments. Third, collaboration with athletic trainers and other professionals improved on-field care and reduced unnecessary hospital transfers. Fourth, paramedics contributed to injury prevention programmes that lowered injury incidence and healthcare costs. Fifth, their involvement at mass gatherings ensured safety, streamlined triage, and reduced pressure on hospitals. Finally, evidence indicates that paramedic-led initiatives are cost-effective, generating both clinical and economic benefits. **Conclusions:** Paramedics play a multifaceted role in athlete care, emergency response, and injury prevention. Strengthening their integration through targeted training, protocol standardization, and equitable resource allocation can improve both athlete safety and system efficiency. Future research should focus on grassroots contexts and the use of paramedic-generated data to inform prevention and policy.

## 1. Introduction

In recent decades, sports medicine has emerged as a critical discipline within the broader field of healthcare, driven by the growing popularity of sports and heightened awareness of athlete safety and performance [1,2]. As competitive and recreational sports continue to expand globally, the need for immediate, specialized medical care at sporting events has become increasingly important [3]. Paramedics, as front-line healthcare providers, have taken on a pivotal role in this evolving landscape, ensuring that athletes receive prompt and appropriate care in high-pressure, often unpredictable environments [4].

In this review, the term ‘paramedic’ refers to healthcare professionals trained to deliver pre-hospital emergency medical care that extends beyond basic life support and includes advanced procedures such as airway management, drug administration, and trauma life support [5]. This definition distinguishes paramedics from other emergency medical personnel, such as emergency medical technicians (EMTs), athletic trainers, or physicians, who may also operate in sporting contexts but with different scopes of practice and legal authority. It is important to note that the role and recognition of paramedics vary significantly across countries; for example, paramedics in Australia, the UK, and Canada often practice at an advanced, autonomous level, whereas in the United States and some European contexts the scope of practice may be narrower or integrated differently into the emergency medical system [6]. Throughout this paper, when the literature uses broader terms such as “emergency medical services personnel” or “first responders,” we clarify how these relate to paramedic functions to avoid confusion in multidisciplinary settings.

With the increasing recognition of sports medicine as a vital component of athlete health and performance, paramedics play a critical role in this field, particularly in managing acute injuries and emergencies at sporting events [7]. Their involvement spans from providing essential life support (BLS) to advanced trauma life support (ATLS), ensuring that athletes receive immediate and effective care at the scene of injury [8,9]. In addition to emergency response, paramedics are increasingly involved in injury prevention programs, working collaboratively with coaches, athletic trainers, and other medical professionals to reduce the incidence and severity of sports-related injuries [10]. This multi-responsibility role requires paramedics to be well-versed in both emergency medical procedures and the specific demands of various sports, highlighting the need for specialized training and ongoing professional development in areas such as advanced trauma and airway management, musculoskeletal injury assessment, concussion recognition, environmental emergencies (e.g., heat-related illness), and interprofessional collaboration with sports medicine teams [8].

Despite the expanding responsibilities of paramedics in sports medicine, the scope and effectiveness of their involvement remain under-researched in many contexts. While roles are reported to vary across sporting levels, types of sports, and geographical regions, the literature offers only fragmented examples rather than systematic comparisons. For instance, elite international events such as the Olympic Games rely on paramedics integrated into large multidisciplinary teams, whereas grassroots competitions often depend on limited pre-hospital coverage [11]. Similarly, the demands of high-contact sports like football or rugby emphasize trauma and concussion management, while endurance events such as marathons or triathlons require expertise in dehydration and environmental emergencies [12]. Regional differences are also evident: in Thailand, motorcycle-based ambulances (motorlances) are deployed to reduce response times at crowded venues, while in Finland and Poland helicopter EMS support has been highlighted in managing mass gatherings [13,14,15]. Understanding these variations more systematically is essential for guiding policy, improving training frameworks, and optimizing medical support systems in athletic environments.

The studies included in this review originate from diverse geographical contexts, including North America, Europe, Asia, and Oceania. This diversity reflects not only differences in healthcare systems and training models, but also variation in how paramedics are integrated into sports medicine teams. Acknowledging these contextual differences is critical for interpreting the findings and for considering how lessons learned in one setting may, or may not, be transferable to others.

Unlike previous narrative or context-specific reports, this review systematically maps the international evidence on paramedics in sports medicine using the Arksey and O’Malley scoping review framework [16]. To our knowledge, this is the first review to explicitly synthesize not only the emergency response role of paramedics but also their preventive functions, training needs, and cost-effectiveness across different sports and regions. By integrating studies from professional and grassroots contexts, this review provides a broader comparative perspective that has not been addressed in earlier literature.

This scoping review provides a comprehensive overview of the impact of paramedics on athlete care, emergency response, and injury prevention in sports medicine. The objectives are to synthesize evidence on their responsibilities, effectiveness, and challenges, while highlighting knowledge gaps that require further research. By aligning with the title and research question, the review clarifies the value of paramedics in diverse sporting contexts and informs best practices, policy development, and training strategies.

## 2. Materials and Methods

### 2.1. Methodological Framework

This scoping review was conducted in line with the Arksey and O’Malley framework, a structured and widely used approach for mapping existing literature on broad and heterogeneous topics. The framework consists of five stages: identifying the research question, conducting a comprehensive search, selecting studies for inclusion, charting the data, and summarizing and reporting the results [16]. Unlike systematic reviews, which focus on evaluating the quality of evidence for narrowly defined interventions, scoping reviews are particularly suited to fields such as sports medicine, where concepts, practices, and contexts vary widely across countries and disciplines. To ensure methodological transparency and completeness, the review was also reported in accordance with the PRISMA-ScR guidelines [17]. This combined approach enabled us to capture the breadth of available evidence, identify key themes and gaps, and provide a reliable basis for informing research, policy, and practice in sports medicine and paramedic services.

### 2.2. Review Question

What is the impact of paramedics on athlete care, emergency response, and injury prevention across different levels of sport (e.g., professional, amateur) and types of events (e.g., mass gatherings vs. local competitions)?

### 2.3. Search Strategy

A systematic search was conducted across PubMed, Scopus, and ScienceDirect, which were selected for their broad coverage of medical, scientific, and multidisciplinary research. The search strategy combined keywords such as “paramedics,” “first responders,” “sports medicine,” “emergency medical services,” and “injury management” to capture the breadth of paramedic roles in sports medicine. The review was reported in accordance with the PRISMA-ScR guidelines to ensure transparency [17]. The search was restricted to articles published in English between January 2000 and December 2023. This time frame was selected to focus on contemporary evidence, as the last decade has seen substantial growth in research on paramedics in sports medicine, particularly in the context of mass sporting events, concussion management, and innovative EMS practices such as motorcycle-based ambulances.

### 2.4. Study Selection and Data Charting

After the initial search, studies were screened using predefined inclusion and exclusion criteria. Studies were eligible if they examined the role of paramedics in sports settings, reported empirical data on their effectiveness in managing sports injuries, or discussed training, preparedness, or collaboration with other medical personnel. Articles published in English between January 2000, and December 2023 were included to capture both early foundational contributions and more contemporary developments in this field. In total, 30 articles were retained for analysis. Studies were excluded if they focused solely on non-sport-related emergencies, were review articles without primary data, or fell outside the defined scope of sports medicine. Data were charted using a standardized template that captured study design, setting, population, interventions, outcomes, and key findings. This systematic approach facilitated the synthesis of information, enabling identification of common themes, significant findings, and knowledge gaps requiring further research.

### 2.5. Study Selection Flow

The database search and screening process are summarized in the PRISMA-ScR flow diagram (Figure 1). A total of 265 records were initially identified through PubMed, Scopus, and ScienceDirect. After removal of 123 duplicates, 142 records were screened by title and abstract, with 24 excluded for lacking a full text or an English version. In total, 118 full-text articles were assessed for eligibility, of which 88 were excluded because they did not specifically focus on paramedics as the study population. Ultimately, 30 studies met the inclusion criteria and were retained for analysis.

## 3. Results

Six overarching themes were identified through synthesis of the findings: (1) response time and accessibility, (2) preparedness and training, (3) effectiveness in accessing athletes at mass gatherings, (4) injury management, (5) cost-effectiveness, and (6) injury prevention. Each of these themes is described in detail in the following subsections.

### 3.1. Overview of Included Studies

A total of 30 studies were included, addressing diverse aspects of paramedic involvement in sports medicine across different settings and sporting events. Of these, 18 were quantitative, 9 qualitative, and 3 used mixed methods. The evidence spans 2000–2023 and covers multiple regions, including North America, Europe, Asia, Oceania, and Africa, thus capturing both high-resource and lower-resource contexts. Together, the studies examine paramedics’ contributions to injury management, preparedness and training, collaboration with other medical professionals, effectiveness during mass gatherings, cost-effectiveness, and preventive strategies.

A summary of study characteristics and key findings is presented in Table 1.

### 3.2. Response Time and Accessibility

Rapid response and accessibility emerged as critical factors in ensuring effective care during sporting events. Several studies demonstrated that optimized transport solutions, such as motorcycle-based ambulances, significantly reduced response times compared to conventional vehicles. For example, Apiratwarakul et al. (2020) [18] reported average access times of 3–4 min for motorlances versus 6–8 min for traditional ambulances, which was associated with lower mortality and improved outcomes. Similar evidence from Poland [20] highlighted the added value of paramedics’ rapid deployment in crowded or urban environments.

Beyond transport innovations, strategic planning and allocation of resources were equally important. Hiltunen et al. showed that careful distribution of EMS personnel and equipment during the World Championships in Finland minimized fatalities and severe injuries [19]. Likewise, Lenjani et al. emphasized the role of preparedness and systematic placement of medical teams in ensuring swift access to injured athletes [24].

Collectively, these findings suggest that improving response times requires both technological solutions and organizational planning. For policymakers and event organizers, integrating such approaches into emergency preparedness plans is essential to safeguard athlete safety and optimize medical outcomes.

### 3.3. Preparedness and Training

Preparedness and training are central to the effectiveness of paramedics in sports medicine. Physical fitness has been repeatedly identified as a key determinant of performance in high-stress, physically demanding environments. Studies from Poland and Australia showed that paramedics with higher fitness levels were better equipped to deliver rapid and competent care, while reduced strength and flexibility among older staff could compromise effectiveness [20,28]. In addition, knowledge gaps in managing complex injuries, such as spine trauma, highlighted the need for ongoing education and interprofessional training [33].

Continuous professional development is therefore essential. Regular training sessions, simulations, and updates on sports injury management protocols ensure that paramedics remain proficient and adaptable to evolving demands [35].

Collaboration with other medical professionals, particularly athletic trainers, further enhances emergency response. Evidence from the USA and UK demonstrated that coordinated teamwork reduced on-field response times, improved injury assessment, and lowered the burden on hospitals [22,25,32].

In summary, physical fitness, continuous training, and collaborative practice form the foundation of paramedic preparedness in sports medicine. Strengthening these dimensions is essential to safeguard athlete health and ensure high-quality emergency response at sporting events [27,38].

### 3.4. Effectiveness in Accessing Athletes in Mass Gathering Sporting Events

Paramedics play a vital role in maintaining safety during large-scale sports events. Evidence from the Netherlands demonstrated that paramedic-led injury prevention programmes can reduce both injury incidence and healthcare costs, highlighting their potential beyond routine emergency response [29,38,43]. Their presence at events also provides reassurance to athletes and spectators, although the preventive effect on risky behaviors requires further empirical validation.

On-site medical capacity is particularly important when hospital transfer may be delayed. Paramedics effectively manage a wide range of conditions, from dehydration and heatstroke to severe trauma, leading to reduced morbidity, fewer hospital transfers, and faster return-to-play times [26,44].

Beyond clinical care, paramedics contribute to pre-event risk assessments, emergency action planning, and medical drills. These activities ensure team readiness and optimize medical resource deployment during events [46]. During crises, their triage skills enable prioritization of severe cases while safely managing minor injuries, maximizing the efficiency of available resources [10]. Furthermore, data collected on injury patterns informs future preparedness and safety planning [30].

### 3.5. Injury Management

Paramedics play a central role in the acute management of sports-related injuries by delivering advanced pre-hospital interventions. Studies from Canada and Japan demonstrated that spinal immobilization, airway management, and other life-preserving procedures performed on-site significantly reduced complications and improved outcomes for injured athletes [31,42,43].

Integration of paramedics into multidisciplinary sports medicine teams has been associated with faster assessments, reduced treatment delays, and shorter recovery times. For example, paramedic involvement in Australia resulted in a 25% reduction in treatment delays and improved recovery trajectories for musculoskeletal injuries [12]. Their presence ensures that athletes receive immediate and appropriate care, reducing the risk of secondary complications.

In addition, paramedics contribute valuable data on injury mechanisms and treatment responses, which can inform future clinical protocols and event planning. This evidence-based approach strengthens both individual athlete care and broader organizational preparedness [39].

### 3.6. Cost-Effectiveness and Injury Prevention

Evidence suggests that paramedic-led injury prevention initiatives are not only clinically beneficial but also economically advantageous. In the UK, Agar demonstrated that such programs reduced emergency department visits by 30% and generated annual savings of approximately £1.2 million [26]. Similarly, U.S. data highlight the broader economic benefits of paramedic involvement, showing reductions in treatment costs and improved efficiency of healthcare resource use [41].

Beyond direct healthcare savings, the visible presence of trained paramedics enhances the reputation and perceived safety of sporting events, which may strengthen participant confidence and encourage sponsor engagement [37,45]. Importantly, these benefits extend beyond professional sport: community and recreational athletes face similar injury risks, and the integration of paramedics at these levels provides both preventive and therapeutic value [45].

Overall, the evidence indicates that investing in paramedic services and prevention programs offers a strong cost–benefit balance, improving athlete safety while reducing the wider economic and healthcare burden of sports-related injuries.

### 3.7. Summary of Findings

Overall, the 30 included studies demonstrate that paramedics play a multifaceted role in sports medicine, extending beyond acute emergency care to encompass preparedness, training, prevention, and system-level efficiency. Across diverse settings, evidence consistently highlights the importance of rapid response times, the necessity of maintaining high levels of physical fitness and continuous education, and the benefits of collaboration with other health professionals. Their presence at mass gatherings improves safety through both clinical and preventive measures, while advanced pre-hospital interventions reduce treatment delays and complications. Finally, paramedic-led programmes show substantial cost-effectiveness, reducing healthcare expenditures and enhancing the reputation and safety of sporting events. Collectively, these findings underline the critical value of paramedics in safeguarding athlete health and strengthening emergency medical systems in sports contexts. A thematic synthesis of the evidence is presented in Table 2, which maps all included studies to the six identified themes and illustrates how the literature collectively supports the multifaceted role of paramedics in sports medicine.

## 4. Discussion

This review highlights the broad contribution of paramedics in sports medicine, extending beyond acute emergency response to encompass preparedness, prevention, and system-level efficiency. A key strength of their involvement is the ability to reduce response times, particularly in crowded or logistically complex venues where traditional ambulances may face delays. Innovations such as motorcycle-based ambulances and helicopters have proven effective in significantly shortening access times and improving survival outcomes [13,30]. However, these solutions remain costly, resource-intensive, and often require specialized training, which may be unfeasible for smaller or resource-limited events. This highlights the importance of context-specific models that balance innovation with financial and operational feasibility.

Physical fitness and readiness constitute another critical factor influencing paramedic effectiveness. Studies consistently link higher physical capacity to performance in demanding environments [20,28]. Nevertheless, questions remain about the sustainability of current fitness standards across an aging workforce or among paramedics with health conditions. Evidence in this area is limited, and further research is needed to ensure long-term workforce capacity and resilience [33].

Equally important is the issue of preparedness and collaboration. Ongoing professional development, simulation training, and interprofessional exercises are essential to maintain competence and adaptability in the evolving field of sports medicine [22,35]. While examples of effective teamwork between paramedics and athletic trainers have been documented, collaboration remains inconsistent across different sporting contexts, resulting in variation in the quality of on-field care [27,42]. Establishing shared training and standardized communication protocols could enhance the reliability and effectiveness of emergency responses.

Despite the evidence of their value, there remains a lack of unified international guidelines or legal frameworks defining the role and responsibilities of paramedics in sports medicine. This regulatory gap contributes to disparities in training, deployment, and integration across countries and levels of sport. Addressing this issue through greater coordination among professional associations, sporting federations, and global health bodies could support the development of harmonized standards, ensuring more equitable and effective emergency care worldwide [37,38].

Evidence also underscores the preventive and economic value of paramedic-led initiatives. Injury prevention programmes have been shown to reduce emergency department visits and generate substantial healthcare savings, with benefits extending to both elite and grassroots sports [29,32]. Nonetheless, such programmes remain underutilized, particularly in community and amateur settings where financial and infrastructural barriers persist. Disparities in access to qualified paramedic care, especially in rural or lower-resource contexts, further underscore the need for targeted investments, including the use of telemedicine and mobile medical units to strengthen regional capacity.

Several gaps in the evidence base were identified. Research rarely quantifies the direct impact of paramedics on clinical outcomes using standardized measures, and few comparative studies exist across countries or levels of sport. Paramedics’ role in concussion care and other complex injuries remains underexplored, as does their capacity to contribute systematically to injury surveillance. While paramedics can collect valuable real-time data during mass gatherings, the integration of these data into larger epidemiological frameworks and decision-making systems has yet to be realized. Furthermore, economic evaluations remain limited, leaving questions about the cost–benefit balance of different deployment models unanswered.

Beyond the immediate context of sports medicine, the integration of paramedics also has broader implications for public health and emergency preparedness [47]. Large-scale sporting events often mirror the challenges of mass gatherings, including crowd management, resource allocation, and the risk of concurrent public health incidents. Strengthening the role of paramedics in these contexts contributes not only to athlete safety but also to community resilience by ensuring rapid, coordinated, and scalable responses to medical emergencies [19,41]. Moreover, lessons learned from sporting environments, such as innovative deployment models, collaborative protocols, and preventive strategies, can inform wider disaster preparedness and health system planning. In this sense, the role of paramedics in sports medicine represents a microcosm of their contribution to population health and highlights the importance of embedding their expertise into broader emergency and public health frameworks [25,38,43,48].

Taken together, the effectiveness of paramedics in sports medicine is not determined by a single factor, but by the interplay of rapid accessibility, physical preparedness, continuous education, interprofessional collaboration, and policy-level support. These elements reinforce one another, forming the basis of resilient and adaptable emergency medical systems in sports environments. More systematic integration of paramedics into sports medicine has the potential to shift practice from a reactive emergency response model to a proactive, data-informed and prevention-oriented approach, ultimately enhancing athlete safety, improving recovery outcomes, and reducing the burden on healthcare systems [42,45].

## 5. Limitations

This scoping review has several limitations. First, only articles published in English were included, which may have excluded relevant research from non-English-speaking countries and introduced language bias. Second, the database search was restricted to PubMed, Scopus, and ScienceDirect, potentially omitting relevant studies indexed in other databases or reported in grey literature. Third, the review focused on peer-reviewed publications, thereby excluding non–peer-reviewed sources such as professional reports, conference proceedings, and operational documents from emergency medical services, which could have provided additional insights into real-world practice.

The included studies also exhibited substantial heterogeneity in design, population, and reporting standards, which complicated direct comparisons and limited the ability to synthesize findings quantitatively. Most studies were observational, context-specific, or small in scale, with limited use of experimental or comparative approaches. In addition, reporting quality varied considerably across countries, reducing the generalizability of results. Although a formal risk of bias assessment was not performed, consistent with scoping review methodology, it is important to acknowledge these methodological limitations and the potential for publication bias.

Finally, the descriptive nature of much of the evidence precluded any quantification of the causal impact of paramedic interventions on clinical outcomes. Consequently, while this review identifies consistent themes and highlights the multifaceted contributions of paramedics in sports medicine, its conclusions should be interpreted with caution and considered as a foundation for future, more rigorous empirical studies.

## 6. Conclusions

This scoping review demonstrates that paramedics play a critical role in sports medicine by enhancing athlete safety, ensuring timely emergency response, and supporting injury prevention. Across diverse settings, evidence highlights three key takeaways.

First, rapid accessibility through innovations such as motorcycle-based ambulances and strategic resource allocation significantly reduces response times and improves outcomes. Second, preparedness and continuous training, combined with interprofessional collaboration, strengthen the ability of paramedics to manage injuries effectively during both routine sports and mass gatherings. Third, paramedic-led prevention initiatives and integration into sports medicine teams not only reduce injury incidence and severity but also generate economic benefits by lowering healthcare costs.

Moving forward, stakeholders should priorities the standardization of protocols, expansion of access in underserved regions, and the use of real-time data systems to optimize decision-making. Strengthening the systemic role of paramedics will shift athlete care from a reactive model to one that is proactive, data-informed, and prevention-oriented.

These findings are consistent across the analytical categories identified in this review, namely response time and accessibility, training and preparation, effectiveness in mass gatherings, injury management, and cost-effectiveness and prevention.

## Figures and Tables

**Figure 1 healthcare-13-02301-f001:**
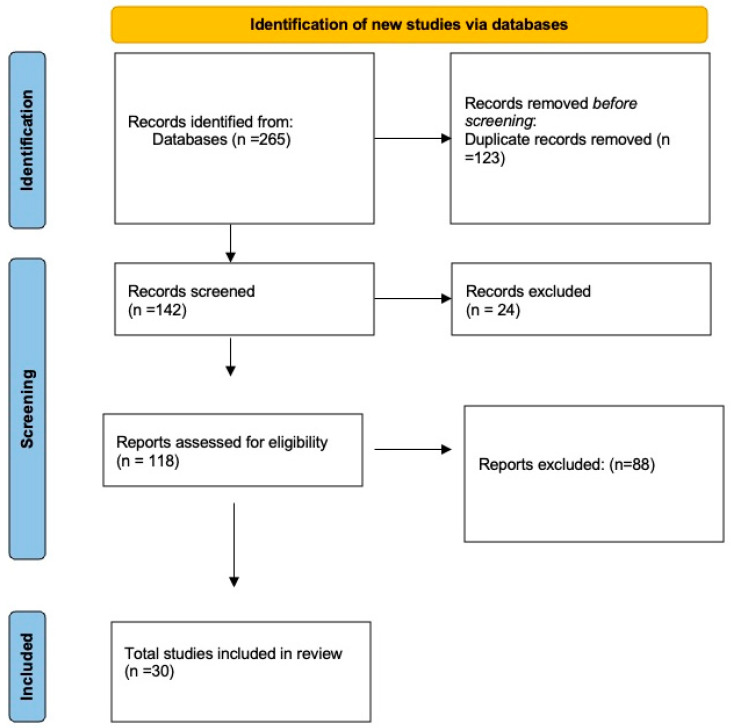
PRISMA-ScR flow diagram of the study selection process.

**Table 1 healthcare-13-02301-t001:** Summary of Included Studies on the Role of Paramedics in Sports Medicine.

Ref	Country	Aim	Design	Population	Key Findings	GRADE
[18]	Thailand	Compare ambulance vs. motorcycle response times	Quantitative	EMS ops at sporting events (2019)	Motorlances halved response times (3–4 vs. 6–8 min), reducing mortality and improving outcomes.	High
[19]	Finland	Assess EMS preparedness at Athletics World Championships	Quantitative	EMS calls, 2005 Games	Adequate planning/resources ensured effective care; fatalities and severe injuries minimised.	High
[20]	Poland	Assess physical activity of HEMS crews	Quantitative	131 HEMS staff, ages 27–59	Fitness levels strongly linked to paramedic preparedness.	High
[21]	Australia	Evaluate safety of on-scene discharge by paramedics	Quantitative	EMS patients, Perth (2013)	Higher risk of adverse events for on-scene discharge vs. ED discharge.	High
[22]	USA	Explore AT–EMS collaboration in injury management	Qualitative	11 EMTs + 6 paramedics	Collaboration improved on-field care, reduced hospital burden.	Moderate
[23]	UK	Examine equestrian injuries with air ambulance support	Quantitative	29 patients, 2008–2009	Paramedics effective in trauma care, supporting physician-led HEMS missions.	High
[24]	Kosovo	Evaluate emergency care at football and other sports	Qualitative	Records from Emergency Clinic (2019)	Training in BLS/ATLS protocols improved injury management and reduced morbidity.	High
[25]	UK	Investigate EMS collaboration at 2012 Olympics	Qualitative	Paramedics + public health staff	Identified leadership and coordination challenges; stressed interagency collaboration.	High
[26]	UK	Document injuries at endurance “Tough Guy” event	Quantitative	251 patients (2006–2007)	1–2% casualty, 5% hospital transfer; trauma/exposure injuries most common.	Moderate
[27]	USA	Assess EMS views on athletic trainers’ roles	Qualitative	115 EMS personnel	Better understanding of ATs could improve protocols and trust.	High
[28]	Australia	Compare paramedic fitness by age/sex	Quantitative	Regional paramedics	Older staff had reduced strength/flexibility; implications for injury response.	High
[29]	The Netherlands	Test paramedic-led soccer injury prevention program	Quantitative	Amateur soccer players	Warm-up program reduced injuries and costs; effective preventive role of paramedics.	High
[30]	Canada	Develop community-based EMS model for multisport event	Quantitative	21,600 athletes, World Masters Games	Low injury rates; community-based EMS model proved safe and practical.	High
[31]	Canada	Evaluate paramedic use of Canadian C-Spine Rule	Quantitative	7 cities, validation data	Highlighted need for further training; immobilisation often appropriate in sports trauma.	High
[32]	USA	Analyse EMS activations for sport injuries	Quantitative	National EMS database (2017–2018)	Most cases required advanced interventions; EMS activation generally justified.	High
[33]	USA	Assess provider knowledge of spine-injured athletes	Quantitative	ATs, EMTs, paramedics	Knowledge gaps common; interprofessional education recommended.	Moderate
[34]	Japan	Describe characteristics of EMS transport for sports injuries	Quantitative	EMS sports patients, Osaka	Injury patterns varied by sport; data useful for prevention planning.	High
[35]	The Netherlands	Evaluate EMS in stadium cardiac emergencies	Qualitative	Patients with cardiac events (2006–2007)	On-site EMS with AEDs lifesaving; recommended for all large venues.	High
[36]	USA	Examine ED physicians’ management of sport concussions	Quantitative	EPs + residents, Michigan	Practice varied; lack of guideline use; highlighted EMS training importance.	High
[37]	USA	Explore EMS role in sudden cardiac arrest at sports fields	Qualitative	Case series, sports SCA patients	On-site EMS presence critical in reducing fatalities.	High
[38]	The Netherlands	Develop EMS guidelines for rave/mass events	Quantitative	Patients at Dutch rave events	Stressed need for paramedics + specialised training for unique risks.	High
[39]	USA	Evaluate ATs’ management of emergencies	Quantitative	EMS directors’ survey	ATs confident, but EMS directors less so; highlighted training needs.	Moderate
[40]	Nigeria	Assess on-site physician/EMS model at sports events	Quantitative	Visitors at mass events	On-site treatment effective, reduced hospital referrals.	High
[41]	USA	Plan EMS for Special Olympics (2015)	Qualitative	Event planning data	Paramedics integral to preparedness and safety.	High
[42]	USA	Explore AT–paramedic teamwork in football injury scenarios	Qualitative	Licensed ATs + EMTs/paramedics	Stronger partnerships improved injury care; training workshops advised.	High
[43]	Australia	Describe in-event emergency facility at marathon	Qualitative	ED staff at Gold Coast Marathon	In-event EMS model effective; prevented hospital transfers.	High
[44]	USA	Document injury patterns at marathon	Quantitative	5 first-aid stations, marathon	Injuries clustered mid/late race; EMS allocation should match demand.	High
[45]	USA	Study EMS transports in school/college athletes	Quantitative	Student-athletes, 23 HS + 25 college	Head/face injuries most frequent; ATs must ensure preparedness.	High
[46]	USA	Test simulation-based EMS/AT teamwork training	Quantitative	ATs in 7 schools	Simulation improved confidence and teamwork; useful for EMS integration.	High

Note: GRADE ratings were extracted from the original studies where available. When not explicitly reported, approximate evidence levels were assigned by the authors based on study design and methodological quality. This classification is intended to guide interpretation rather than serve as a formal risk of bias assessment.

**Table 2 healthcare-13-02301-t002:** Mapping of Included Studies to Identified Themes.

Theme	Key Focus	Representative Findings	References
Response time & accessibility	Speed and efficiency of EMS response	Motorlances halved response times; strategic allocation reduced fatalities.	[18,19,21,24,26,44]
Preparedness & training	Fitness, ongoing education, teamwork	Higher fitness improved performance; gaps in spine-injury knowledge; AT–EMS collaboration improved on-field response.	[20,22,25,27,28,33,42,46]
Effectiveness in mass gatherings	EMS role at large-scale events	On-site EMS reduced hospital transfers; ensured acute care and safety in mass events.	[23,24,25,29,30,38,41,43]
Injury management	Acute pre-hospital interventions	Spinal immobilization, airway management; reduced treatment delays and complications.	[21,31,34,36,37,39]
Cost-effectiveness	Economic value of EMS services	Prevention programs reduced ED visits and costs; savings up to £1.2 m annually.	[26,29,41,45]
Injury prevention	Proactive safety strategies	Warm-up programs, risk assessments, education reduced injuries and guided planning.	[29,32,35,40,45,46]

Note: Not all included studies are listed under a single theme in Table 2. Some studies addressed multiple themes, while others provided broader contextual insights that did not align exclusively with one analytical category. References shown here illustrate representative examples of how the evidence base mapped onto the six themes.

## Data Availability

The datasets used and/or analyzed during the current study are available from the corresponding author on reasonable request.

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
