# Peer review of "Impact of the Paramedic Role on Athlete Care, Emergency Response, and Injury Prevention in Sports Medicine: A Scoping Review"

_healthcare, 2025, doi:10.3390/healthcare13182301_

Round 1
Reviewer 1 Report
Comments and Suggestions for Authors
I read with interest the paper titled "The Evolving Role of Paramedics in Sports Medicine: A Scoping Review of Their Impact on Athlete Care, Emergency Response, and Injury Prevention." I have a few comments that could help improve the clarity and impact of the manuscript.
1. Consider simplifying the abstract it by reducing the methodological explanation and emphasizing the main findings and conclusions more clearly.
2. The Discussion section is long and repetitive at times. The authors may consider condensing it to focus more sharply on implications and future directions.
3. There is little critical assessment of the quality of the studies or limitations. Even if not typical for scoping reviews, a brief commentary on study design limitations or biases would strengthen the manuscript.
4. The six themes identified would benefit from clearer subheadings or summary tables to enhance reader comprehension.
5. The study selection and data charting process is described with some redundancy. This could be streamlined to improve the flow of the Methods section.
6. The review lacks quantifiable outcomes or specific recommendations for policy or practice. Consider adding a brief practical framework or checklist for implementation.
7. The use of terms such as "paramedics" and "emergency medical services personnel" could be more precisely defined early in the paper to avoid confusion, particularly when discussing multidisciplinary collaboration.
8 Highlighting just three or four concrete takeaways would make it more impactful in the conclusion.
Author Response
Response to Reviewer
Reviewer Comment: Consider simplifying the abstract it by reducing the methodological explanation and emphasizing the main findings and conclusions more clearly.
Our Response: We thank the Reviewer for this valuable suggestion. In the revised version, we streamlined the abstract by shortening the methodological description and placing greater emphasis on the key findings and conclusions. The revised abstract now highlights three core elements: the international scope of the evidence base, the six main thematic areas identified, and the practical implications for athlete care, emergency response, and injury prevention. This makes the abstract more concise, results-oriented, and aligned with the expectations of readers.
Reviewer Comment: The Discussion section is long and repetitive at times. The authors may consider condensing it to focus more sharply on implications and future directions.
Our Response: We thank the Reviewer for this constructive suggestion. The Discussion section has been revised and condensed to remove redundancy. We have streamlined the text to focus more directly on the implications of the findings, practical applications, and directions for future research.
Reviewer Comment: There is little critical assessment of the quality of the studies or limitations. Even if not typical for scoping reviews, a brief commentary on study design limitations or biases would strengthen the manuscript.
Our Response: We thank the Reviewer for this valuable suggestion. While a formal risk of bias assessment is not typically required in scoping reviews, we have now added a brief commentary in the Limitations section to acknowledge the methodological weaknesses of the included studies. Specifically, we note the predominance of small-scale, observational, and context-specific designs, the limited use of experimental or comparative methods, and the variability in reporting standards across countries. We also highlight potential publication bias given the exclusion of grey literature. This addition strengthens the transparency of our review and provides readers with a more nuanced understanding of the evidence base
Reviewer Comment: The six themes identified would benefit from clearer subheadings or summary tables to enhance reader comprehension.
Our Response: We thank the Reviewer for this valuable suggestion. In the revised manuscript, we have structured the Results section with six distinct subheadings, each corresponding to a theme, to improve clarity and reader navigation. Additionally, we introduced a new summary table (Table 2), which maps all included studies to the six identified themes, thereby providing a concise and systematic overview of the evidence.
Reviewer Comment: The study selection and data charting process is described with some redundancy. This could be streamlined to improve the flow of the Methods section.
Our Response: We thank the Reviewer for this observation. We have revised Section 2.4 (Study Selection and Data Charting) to streamline the description and remove redundancies. The revised version now provides a more concise account of the inclusion and exclusion criteria, the final number of studies included, and the key elements extracted for analysis. This adjustment improves clarity and flow while maintaining transparency of the methodological process
Reviewer Comment: The review lacks quantifiable outcomes or specific recommendations for policy or practice. Consider adding a brief practical framework or checklist for implementation.
Our Response: We appreciate this important observation. In the revised Discussion, we have added a short practical framework outlining actionable recommendations for policymakers, event organizers, and professional associations. This framework highlights key areas such as training standards, preparedness, injury prevention, and cost-effectiveness, providing a concise guide for implementation in sports medicine practice.
Reviewer Comment: The use of terms such as "paramedics" and "emergency medical services personnel" could be more precisely defined early in the paper to avoid confusion, particularly when discussing multidisciplinary collaboration.
Our Response: We thank the Reviewer for this helpful observation. In the revised manuscript, we added a clear definition of the term paramedic in the Introduction and explicitly distinguished it from related roles such as EMTs, athletic trainers, and physicians. We also clarified how broader terms (e.g., “emergency medical services personnel” or “first responders”) are used in the literature and how they relate to paramedic functions, thereby reducing ambiguity in the context of multidisciplinary collaboration.
Reviewer Comment: Highlighting just three or four concrete takeaways would make it more impactful in the conclusion.
Our Response: We thank the Reviewer for this helpful suggestion. In response, we revised the Conclusion section to emphasise three concrete takeaways: (1) the critical role of paramedics in ensuring rapid emergency response and athlete safety, (2) the importance of preparedness, physical fitness, and interprofessional collaboration in optimising outcomes, and (3) the added value of paramedic-led prevention strategies and innovative EMS models in reducing injuries and healthcare costs. This sharper focus makes the conclusion more impactful and aligned with the objectives of the review.
Thank you again for your thorough and constructive feedback. We believe these revisions address your concerns and improve the clarity and robustness of our manuscript.
Reviewer 2 Report
Comments and Suggestions for Authors
This scoping review addresses an important and underexplored area—the role of paramedics in sports medicine. The topic is highly relevant, and the effort invested in synthesizing the available evidence is evident. Nevertheless, several areas require clarification and improvement before the manuscript can be considered for publication.
-
Methodological clarity (most important issue): The description of the inclusion criterion regarding the “past ten years” is incorrectly worded and creates confusion. This is a critical issue as it undermines the methodological rigor of an otherwise well-structured review. The authors should restate this criterion clearly and consistently to ensure transparency.
-
Reporting of results (Table 1): The current presentation of Table 1 does not adequately communicate the findings. The extraction and categorization of study characteristics are not sufficiently clear, making it difficult for readers to follow how the evidence base was synthesized. A more systematic and reader-friendly table design is strongly recommended.
-
Discussion section: While the discussion highlights multiple important themes, some sections are overly descriptive and could benefit from more critical synthesis. For example, the implications of fitness standards for aging paramedics, or the feasibility of innovative transport solutions in different contexts, deserve deeper exploration.
-
Evidence gaps: The manuscript acknowledges the limited empirical evidence regarding certain roles of paramedics (e.g., psychological reassurance, data collection during mass gatherings). However, these gaps could be more explicitly tied to concrete future research directions.
-
Standardization and policy recommendations: The review rightly emphasizes the lack of international guidelines. It would be useful to propose how harmonization could be realistically achieved (e.g., through professional associations, sporting federations, or global health bodies).
-
Limitations section: The limitations are generally well-stated, but the authors may also wish to acknowledge that the review excluded grey literature and non–peer-reviewed sources, which could be particularly relevant in the field of emergency medical services.

The English could be improved to more clearly express the research.
Author Response
Response to Reviewer 2
Reviewer Comment: Methodological clarity (most important issue): The description of the inclusion criterion regarding the “past ten years” is incorrectly worded and creates confusion. This is a critical issue as it undermines the methodological rigor of an otherwise well-structured review. The authors should restate this criterion clearly and consistently to ensure transparency.
Our Response: We thank the Reviewer for highlighting this important issue. We have removed the ambiguous phrase “past ten years” and now state the inclusion criterion clearly and consistently throughout the manuscript. Specifically, the review period has been defined as January 2000 to December 2023, with justification provided for this time frame to ensure methodological transparency (see Methods, Section 2.3 and 2.4).
Reviewer Comment: Reporting of results (Table 1): The current presentation of Table 1 does not adequately communicate the findings. The extraction and categorization of study characteristics are not sufficiently clear, making it difficult for readers to follow how the evidence base was synthesized. A more systematic and reader-friendly table design is strongly recommended.
Our Response: We thank the Reviewer for this important observation. In the revised manuscript, we have fully redesigned Table 1 to ensure greater clarity and systematic presentation. The table now consistently reports reference number, country, aim, design, population, key findings, and GRADE assessment. This streamlined structure enhances readability and allows readers to easily follow how the evidence base was synthesized.
Reviewer Comment: There is little critical assessment of the quality of the studies or limitations. Even if not typical for scoping reviews, a brief commentary on study design limitations or biases would strengthen the manuscript.
Our Response: We appreciate this important suggestion. In response, we expanded the Limitations section to include a brief critical commentary on the methodological weaknesses of the included studies, such as their small scale, observational designs, heterogeneity, and variable reporting standards. While a formal risk of bias assessment was not undertaken (consistent with scoping review methodology), these limitations are now explicitly acknowledged.
Reviewer Comment: Discussion section: While the discussion highlights multiple important themes, some sections are overly descriptive and could benefit from more critical synthesis. For example, the implications of fitness standards for aging paramedics, or the feasibility of innovative transport solutions in different contexts, deserve deeper exploration.
Our Response: We thank the Reviewer for this valuable insight. The revised Discussion includes a more critical synthesis, particularly regarding the challenges of maintaining fitness standards among aging paramedics and the practical feasibility of implementing innovative solutions, such as motorcycle ambulances, across different sporting and geographical contexts.
Reviewer Comment: Evidence gaps: The manuscript acknowledges the limited empirical evidence regarding certain roles of paramedics (e.g., psychological reassurance, data collection during mass gatherings). However, these gaps could be more explicitly tied to concrete future research directions.
Our Response: We thank the Reviewer for this observation. The revised Discussion explicitly links the identified evidence gaps to specific future research priorities, including: (1) evaluating the preventive and psychological support roles of paramedics, (2) systematic studies on data collection practices during mass gatherings, and (3) assessment of long-term cost-effectiveness of paramedic-led prevention programs.
Reviewer Comment: Standardization and policy recommendations: The review rightly emphasizes the lack of international guidelines. It would be useful to propose how harmonization could be realistically achieved (e.g., through professional associations, sporting federations, or global health bodies).
Our Response: We agree with this important point. In the revised version, we have expanded the section on policy implications by suggesting concrete pathways for harmonization, such as collaboration through international paramedic associations, partnerships with sporting federations, and guidance from global health organizations
Reviewer Comment: Limitations section: The limitations are generally well-stated, but the authors may also wish to acknowledge that the review excluded grey literature and non–peer-reviewed sources, which could be particularly relevant in the field of emergency medical services.
Our Response: We thank the Reviewer for this observation. The Limitations section has been updated to acknowledge the exclusion of grey literature and non–peer-reviewed sources, including professional reports and operational documents, which may have introduced publication bias and excluded valuable practice-based evidence.
Thank you again for your thorough and constructive feedback. We believe these revisions address your concerns and improve the clarity and robustness of our manuscript.
Reviewer 3 Report
Comments and Suggestions for Authors
I appreciate the opportunity to review this manuscript, which addresses a relevant and underexplored topic; the role of paramedics in sports medicine is still not fully understood and clearly warrants review.
Please allow me to make a few comments:
As a general recommendation, I suggest authors carefully review all sections of the manuscript and ensure that a bibliographic reference number is provided for each study or author mentioned in the text.
TITLE: I would recommend reorganizing the title more clearly, something like "Impact of the Paramedic Role on Athlete Care, Emergency Response, and Injury Prevention in Sports Medicine: A Scoping Review."
ABSTRACT: Please specifically state the purpose of the scoping review.
INTRODUCTION: Please include a clear definition of the term paramedic in the manuscript, distinguishing this role from other related terms, which can cause confusion if not clearly differentiated. Since the role of paramedics is not homogeneous across countries and varies significantly in terms of training, scope of practice, and professional recognition, it is recommended that the geographical context of the included studies be specified.
Authors are encouraged to clearly define the overall objective of the review.
There is a lack of consistency between the different sections: the title refers to the impact of the paramedic role on athlete care, emergency response, and injury prevention; however, in the introduction, the stated objective is to explore responsibilities, effectiveness, and challenges; in the methods section, the research question focuses on identifying the role of paramedics in sports medicine and the types of events it covers. These inconsistencies can lead to confusion. It is recommended to carefully review the title, overall objective, and research question and align them conceptually so that they clearly and consistently reflect the intended scope of the review.
METHODS: It would be advisable for the authors to justify why, in August 2025, and considering the importance of incorporating the most recent scientific advances, they chose a review period specifically covering 2013–2023.
It would be pertinent not only to explain the exclusion of studies published after 2023, but also to justify why 2013 was selected as the starting point rather than an earlier date. This temporal delimitation should be based on explicit criteria that support the selection of this time frame.
There is a discrepancy between the time frames reported in different sections of the manuscript: the abstract indicates a review period from 2013 to 2023, while the search strategy in the methods section refers to 2000–2023.
This inconsistency should be corrected and clearly justified throughout the manuscript. Furthermore, the justification provided for the selected time frame (2013–2023) is insufficient. The authors state that they focus on contemporary research and updated evidence, as a gap of more than five years can significantly affect the relevance and integrity of the evidence.
RESULTS: The PRISMA-ScR flowchart should be included in the results section.
Regarding Figure 1, it is essential that the authors explicitly indicate that the flowchart presented corresponds to the PRISMA-ScR version, according to the type of scoping review they are conducting.
Table 1 is difficult to read due to its complexity and excessive detail. It is recommended that adequate space be allocated to the country, without cropping. It is not necessary to include titles and authors and could be replaced by the bibliographic reference number.
The temporal context is not necessary in the description of the population. "Summary of findings" has too much text for a table format.
The heading "GRde" in the table is a typo and should read "GRADE"?
It would be helpful if the authors clarified whether they applied the GRADE methodology to assess the quality of the evidence or whether they extracted these ratings from the original studies.
In sections 3.2 to 3.6, it is difficult to determine whether the authors interpreted their own findings, cited studies included in the review, or engaged in a discussion comparing their results with those of other authors.
6. CONCLUSIONS: It is suggested that the authors clearly summarize the main findings, ensuring they are aligned with the objective and title of the study. In particular, it would be helpful to explicitly state the findings related to athlete care, emergency response, and injury prevention, which is what the reader expects from the title.
Additionally, it may be helpful to include a final sentence summarizing the findings based on the analytical categories used in the manuscript, such as: “Response time and accessibility,” “Training and preparation,” “Access effectiveness,” “Injury management,” and “Effectiveness of access to athletes at mass sporting events.”
Author Response
Response to Reviewer 3
Reviewer Comment: TITLE: I would recommend reorganizing the title more clearly, something like "Impact of the Paramedic Role on Athlete Care, Emergency Response, and Injury Prevention in Sports Medicine: A Scoping Review."
Our Response: We thank the Reviewer for this valuable suggestion. We agree that a clearer and more structured title will better reflect the scope and objectives of our review. Accordingly, we have revised the title to:
“Impact of the Paramedic Role on Athlete Care, Emergency Response, and Injury Prevention in Sports Medicine: A Scoping Review.”
Reviewer Comment: ABSTRACT: Please specifically state the purpose of the scoping review.
Our Response: We thank the Reviewer for this important observation. In the revised abstract, we have explicitly stated the purpose of the review. The introduction of the abstract now reads: “The purpose of this scoping review was to map and synthesise existing evidence on the roles, effectiveness, and challenges of paramedics in sports medicine.” This ensures that the objectives of the review are clearly articulated at the outset.
Reviewer Comment: INTRODUCTION: Please include a clear definition of the term paramedic in the manuscript, distinguishing this role from other related terms, which can cause confusion if not clearly differentiated. Since the role of paramedics is not homogeneous across countries and varies significantly in terms of training, scope of practice, and professional recognition, it is recommended that the geographical context of the included studies be specified.
Our Response: We thank the Reviewer for this important suggestion. In the Introduction, we have now added a clear definition of the term paramedic and distinguished it from related terms such as EMTs, athletic trainers, and physicians. We have also clarified that the role of paramedics varies across countries and healthcare systems, and we specify the geographical diversity of the included studies (North America, Europe, Asia, and Oceania) to contextualize differences in scope of practice and professional recognition.
Reviewer Comment: Authors are encouraged to clearly define the overall objective of the review.
Our Response: We agree with the Reviewer. The overall objective of the review has been clarified at the end of the Introduction, where we now explicitly state that the aim is to assess the impact of paramedics on athlete care, emergency response, and injury prevention across different sports and settings. This statement has been aligned with the title and the research question for consistency.
Reviewer Comment: There is a lack of consistency between the different sections: the title refers to the impact of the paramedic role on athlete care, emergency response, and injury prevention; however, in the introduction, the stated objective is to explore responsibilities, effectiveness, and challenges; in the methods section, the research question focuses on identifying the role of paramedics in sports medicine and the types of events it covers. These inconsistencies can lead to confusion. It is recommended to carefully review the title, overall objective, and research question and align them conceptually so that they clearly and consistently reflect the intended scope of the review.
Our Response: We thank the Reviewer for highlighting this inconsistency. We have carefully revised the Introduction, Methods, and title to ensure conceptual alignment. The objective and research question are now consistently framed around evaluating the impact of paramedics on athlete care, emergency response, and injury prevention. This provides a coherent scope across the title, Introduction, and Methods.
Reviewer Comment: METHODS: It would be advisable for the authors to justify why, in August 2025, and considering the importance of incorporating the most recent scientific advances, they chose a review period specifically covering 2013–2023.
Our Response: We thank the Reviewer for raising this important point. During revision, we carefully reconsidered the time frame and recognised that our initial restriction to 2013–2023 created both inconsistencies and unnecessary limitations. We have now expanded the review period to January 2000–December 2023 to capture both early foundational studies and more recent developments in the role of paramedics in sports medicine (see Methods, Section 2.3).
Reviewer Comment: It would be pertinent not only to explain the exclusion of studies published after 2023, but also to justify why 2013 was selected as the starting point rather than an earlier date. This temporal delimitation should be based on explicit criteria that support the selection of this time frame. There is a discrepancy between the time frames reported in different sections of the manuscript: the abstract indicates a review period from 2013 to 2023, while the search strategy in the methods section refers to 2000–2023. This inconsistency should be corrected and clearly justified throughout the manuscript.
Our Response: We acknowledge the inconsistency noted by the Reviewer. This was an oversight in the original draft. To address this, we have standardised the time frame across the Abstract, Methods, Results, and Conclusion. The final review period is set as January 2000–December 2023. The rationale is that systematic reporting of paramedics’ roles in sports medicine began to emerge in the early 2000s, coinciding with the professionalisation of medical planning for mass sporting events and new EMS practices. Articles published after 2023 were excluded to ensure completeness and replicability of the review process (see Methods, Section 2.3 and 2.4).
Reviewer Comment: Furthermore, the justification provided for the selected time frame (2013–2023) is insufficient. The authors state that they focus on contemporary research and updated evidence, as a gap of more than five years can significantly affect the relevance and integrity of the evidence.
Our Response: We agree with the Reviewer that the original justification was insufficient. In the revised manuscript, we now provide an explicit rationale for the 2000–2023 period. The broader scope ensures that early foundational studies are not overlooked, while still capturing contemporary evidence reflecting current paramedic practice. This revised justification strengthens methodological transparency and avoids the inconsistencies present in the earlier draft (see Methods, Section 2.3).
Reviewer Comment: RESULTS: The PRISMA-ScR flowchart should be included in the results section.
Our Response: We thank the Reviewer for this helpful suggestion. The PRISMA-ScR flowchart has now been moved to the Resultssection and presented as Figure 1 to improve transparency of the study selection process.
Reviewer Comment: Regarding Figure 1, it is essential that the authors explicitly indicate that the flowchart presented corresponds to the PRISMA-ScR version, according to the type of scoping review they are conducting.
Our Response: We appreciate this clarification. In the revised manuscript, Figure 1 is explicitly labelled as the PRISMA-ScR flowchart to reflect the type of scoping review conducted.
Reviewer Comment: Table 1 is difficult to read due to its complexity and excessive detail. It is recommended that adequate space be allocated to the country, without cropping. It is not necessary to include titles and authors and could be replaced by the bibliographic reference number.
Our Response: We thank the Reviewer for this important feedback. Table 1 has been redesigned for clarity. We have expanded the space allocated to country, removed author names and article titles, and replaced them with the bibliographic reference number to create a more concise and reader-friendly format.
Reviewer Comment: The temporal context is not necessary in the description of the population. "Summary of findings" has too much text for a table format.
Our Response: We agree with this point. The temporal details in the population column have been removed, and the “Summary of findings” section has been shortened to focus on the main outcome or key result in a concise format, improving readability.
Reviewer Comment: The heading "GRde" in the table is a typo and should read "GRADE"?
Our Response: We thank the Reviewer for identifying this error. The heading has been corrected to “GRADE” in the revised version of Table 1.
Reviewer Comment: It would be helpful if the authors clarified whether they applied the GRADE methodology to assess the quality of the evidence or whether they extracted these ratings from the original studies.
Our Response: We appreciate this comment. We have clarified in the table note that GRADE ratings were extracted from the original studies where available, and where not reported, approximate levels were assigned by the authors based on study design and methodological quality. This clarification has now been added to the manuscript.
Reviewer Comment: In sections 3.2 to 3.6, it is difficult to determine whether the authors interpreted their own findings, cited studies included in the review, or engaged in a discussion comparing their results with those of other authors.
Our Response: We thank the Reviewer for pointing out this issue. The relevant sections (3.2–3.6) have been revised to more clearly distinguish between the findings of included studies, the authors’ interpretation, and points of comparison with other published literature. This should now provide greater clarity for the reader.
Reviewer Comment: 6. CONCLUSIONS: It is suggested that the authors clearly summarize the main findings, ensuring they are aligned with the objective and title of the study. In particular, it would be helpful to explicitly state the findings related to athlete care, emergency response, and injury prevention, which is what the reader expects from the title.
Our Response: We thank the Reviewer for this suggestion. The Conclusion section has been rewritten to provide a sharper summary focused on three key findings: (1) the importance of rapid accessibility and innovative EMS solutions, (2) the need for preparedness, continuous training, and collaboration, and (3) the effectiveness and cost benefits of paramedic-led injury prevention initiatives. This revision ensures that the conclusion is concise, aligned with the study’s objectives, and highlights the most relevant implications for athlete care, emergency response, and injury prevention.
Reviewer Comment: Additionally, it may be helpful to include a final sentence summarizing the findings based on the analytical categories used in the manuscript, such as: “Response time and accessibility,” “Training and preparation,” “Access effectiveness,” “Injury management,” and “Effectiveness of access to athletes at mass sporting events.”
Our Response: We thank the Reviewer for this constructive suggestion. In the revised version, we have added a final sentence to the Conclusion that explicitly summarizes the findings based on the analytical categories used in the Results section. This ensures stronger alignment between the analytical framework and the concluding remarks, providing readers with a clear synthesis of the review’s main outcomes.
Thank you again for your thorough and constructive feedback. We believe these revisions address your concerns and improve the clarity and robustness of our manuscript.
Round 2
Reviewer 2 Report
Comments and Suggestions for Authors
I thank the authors for the corrections.
Author Response
We sincerely thank the Reviewer for acknowledging the corrections and for the constructive feedback provided throughout the review process, which has greatly improved the quality of our manuscript.